

**Brown carbon in fine particles in four typical cities in North-**
**west China during wintertime: coupling optical properties**
**with chemical processes**
Miao Zhong[a,b], Jianzhong Xu[a,*], Huiqin Wang[c], Li Gao[d], Haixia Zhu[e], Lixiang Zhai[a,b],
Xinghua Zhang[a], Wenhui Zhao[a,b]
[a]State Key Laboratory of Cryospheric Sciences, Northwest Institute of Eco-Environment and Resources,
Chinese Academy of Sciences, Lanzhou 730000, China
[b]University of Chinese Academy of Sciences, Beijing 100049, China
[c]Institute of Desert Meteorology, China Meteorological Administration, Urumqi 830002, China
[d]College of Resources and Environment, Ningxia University, Yinchuan 750021, China
[e]Key Laboratory of Comprehensive and Highly Efficient Utilization of Salt Lake Resources, Qinghai
Institute of Salt Lakes, Chinese Academy of Science, Xining, Qinghai 810008, China
*Corresponding to*: Jianzhong Xu (jzxu@lzb.ac.cn)





**Abstract.** Brown carbon (BrC) aerosol could impact atmospheric radiative forcing and play a crucial role in atmospheric photochemistry. Most previous studies have predominantly focused on bulk optical properties of ambient BrC from biomass burning emitted primary or secondary BrC aerosol. Few studies have focused on fossil-fuel-influenced BrC aerosol, especially coal combustion emissions. In this study, fine particulate matter ($PM_{2.5}$) filter samples were collected synchronously in four capital cities of North-west China during the winter season (December 2019–January 2020): Lanzhou (LZ), Xining (XN), Yin-chuan (YC), and Urumqi (UR), which are represented as energy-producing and heavy manufacturing cities in China. The aim of the study was to explore the optical properties, sources, and chemical processes of water-soluble BrC (WS-BrC). The average mass absorption efficiency at 365 nm (MAE365) of WS-BrC at these four cities were $1.24 \pm 0.19$ m²/g (XN), $1.19 \pm 0.12$ m²/g (LZ), $1.07 \pm 0.23$ m²/g (YC), and $0.78 \pm 0.16$ m²/g (UR), respectively. The properties of WS-BrC were investigated by an acid-base titration experiment. The MAE365 values in all cities increased with increasing pH values (2–11), while the fluorescent intensities of water extracts fluctuated with corresponding pH values, being stronger at higher acidic and basic conditions. The WS-BrC at YC and LZ were the two most sensitive sites to pH variation, suggesting the important contribution of acid/base functional groups. Furthermore, significant photo-enhancement (LZ) or photo-bleaching (YC and UR) phenomena based on coupling bulk chemical properties with light absorption properties were observed in different cities, indicating their sources and/or chemical processes were different among each other.

The sources and chemical processes of WS-BrC were further explored by the combination of parallel factor analysis (PARAFAC) on excitation-emission matrix of WS-BrC and positive matrix factorization analysis (PMF) on high-resolution mass spectra of water-soluble organic aerosol (OA). Six PARAFAC components including three humic-like substances (LO-HULIS, HO-HULIS1, and HO-HULIS2), two protein-like (PLS) or phenol-like substances, and one undefined substance were obtained. Four PMF factors including a water-soluble primary OA (WS-POA), a less oxidized oxygenated OA that associated with coal combustion-induced WSOA (LO-OOA), and two highly oxidized oxygenated OAs respectively from photochemical oxidation and aqueous-phase oxidation transformations (HO-OOA1 and HO-OOA2) were identified. WS-POA was the most important source of light absorption accounting for 30%–60% based on multiple linear regression model and was significantly correlated with PLS and LO-HULIS



45 components. The loss of light absorption of WS-POA is accomplished by conversion to LO-OOA and

46 HO-OOAs through photo- or aqueous reactions, where HO-OOAs were significantly correlated with

47 HO-HULIS component. The potential precursors and reaction pathways for WS-BrC in each city are

48 proposed.



**1. Introduction**

Brown carbon (BrC) is a certain fraction of organic aerosols that absorb lights in the ultraviolet and visible (UV−Vis) ranges (Andreae and Gelencsér, 2006; Laskin et al., 2015). The light absorption of BrC displays a strong wavelength-dependence which can be characterized by higher value (≥ 2) of the absorption Ångström exponent (AAE) (Laskin et al., 2015). The significant effects of BrC on climate and atmospheric chemistry have been characterized previously. Wang et al. (2018) estimate the global mean absorption direct radiative effect (DRE) of BrC is +0.048W/m² using the GEOS-Chem chemical transport model. The absorption of solar radiation due to BrC can also affect the formation of ozone and radicals of •OH/•HO$_2$ and corresponding atmospheric chemical reactions (Mok et al., 2016; Baylon et al., 2018).

Biomass combustion is a major global source of primary BrC, as biomass is widely used for residential heating, cooking, and is also produced during wildfires (Washenfelder et al., 2015; Lin et al., 2017; Zeng et al., 2020). In addition to primary sources, secondary BrC is formed through various reaction pathways related to biomass burning, such as aqueous oxidation of phenolic compounds or gas-phase photo-oxidation of aromatic volatile organic compounds (VOCs) (Lin et al., 2015; Vidovic et al., 2018; Liu et al., 2019; Vidović et al., 2020). The chemical compositions and light absorption of BrC can vary significantly due to atmospheric aging. For instance, BrC can photobleaching through photolysis reactions in the presence of •OH radical and O$_3$, or darken via the formation of nitrated organic compounds (Lin et al., 2015; Zhao et al., 2015; Moise et al., 2015; Li et al., 2020a). Furthermore, atmospheric conditions such as changes in pH, air temperature, and relative humidity can affect the light absorption characteristics of BrC (Song et al., 2013; Moise et al., 2015; Phillips et al., 2017; Qin et al., 2020).

Various optical instruments are used to determine the light absorption of BrC. These instruments include direct measurement of airborne aerosol (e.g., particle soot absorption photometer, photo-acoustic spectroscopy, and cavity ring-down spectroscopy) (Laskin et al., 2015) or offline measurement of aerosol extracts by UV-Vis spectrometer (Hecobian et al., 2010). Excitation-emission matrix spectroscopy (EEMs) was recently used to reveal similar chemical structures and photochemical features, as well as



to trace the sources of BrC chromophores (Chen et al., 2016; Tang et al., 2020). A few recent studies
have characterized BrC compounds by combining high-resolution mass spectrometry and UV-vis spec-
troscopy (Lin et al., 2015; Lin et al., 2016; Lin et al., 2018; Wang et al., 2019; Huang et al., 2020; Ni et
al., 2021), which facilitates the assessment of the chemical composition of BrC chromophores at the
molecular level. For instance, Xu et al. (2020b) discovered that the light absorption of water-soluble
organic carbon (WSOC) in the southern regions of TP is higher than that in the northern regions of TP,
as well as a considerable variation in the molecular composition of WSOC at two regions.

In China, coal is still a primary source of energy due to its extensive use in coal-fired power plants,
industrial steam boilers, and central heating during winter (Zhang et al., 2008; Yang et al., 2020b). It is
particularly prevalent in Western China, where numerous industrial bases have been established since
the Development of China's Western Regions strategy in the 2000s. In recent years, cities in Northwest
China have experienced more severe air pollution due to rapid economic development and intensive
anthropogenic activities, especially in the capital cities of this region. Some recent studies have identified
coal combustion as a significant source of BrC in Northwest China (Tan et al., 2016; Chen et al., 2021;
Zhang et al., 2021a). Compared with BrC resulting from biomass burning, the optical and chemical prop-
erties of BrC emitted from coal combustion have not been well-characterized.

The aim of this study is to characterize the optical properties of water-soluble BrC (WS-BrC) by collect-
ing PM$_{2.5}$ filter samples from four capital cities in Northwest China and analyzing them using a suite of
instruments. In particular, the study focused on the contribution of primary and secondary sources of
atmospheric chromophores and the related chemical processes. This was achieved by combining data
from Excitation-Emission Matrices (EEMs) and High-Resolution Aerosol Mass Spectrometry (HR-
AMS).

**2. Methods**
2.1. Filter samples collection at the four cities

PM$_{2.5}$ filter samples were collected synchronously from four capital cities in Northwest China, namely



Yinchuan (YC), Xining (XN), Urumqi (UR), and Lanzhou (LZ), from 5th December 2019 to 20th Janu-
ary 2020 (Figure 1). A total of 14 filter samples were collected from each city twice a week. The sam-
pling site, located in the cultural and educational districts of each city, were on the rooftops of buildings
above 20 meters from the ground and free from significant pollution sources. A medium volume $PM_{2.5}$
air sampler (Laoying Ltd., Qingdao, model 2030) with a flow rate of 100 L/min was used for collecting
samples in YC (sample IDs: 1−14), while low volume PM2.5 air samplers (Wuhan Tianhong Instrument
Co. LTD, TH-16E) with a flow rate of 16.7 L/min and BGI PQ200 low-flow rate aerosol samplers (Mesa
Labs, Butler, NJ, USA) were used to collect samples in XN and UR (sample IDs: 15−28 and IDs: 29–
42), and LZ (sample IDs: 43−56), respectively. Quartz filters of 90-nm diameter (Whatman, UK) were
used for sampling in YC, while in the other three cities, quartz fiber filters of 47-mm diameter (PALL
Life Sciences, USA) were used. Before sampling, quartz filters were roasted at 550 °C for five hours to
remove carbonaceous particles. Blank filter samples were obtained from each site by leaving the filters
in the sampler for ten minutes without sampling. Each filter was wrapped in aluminum foil and frozen at
−20 °C until analysis. Daily average concentration of $PM_{2.5}$, $SO_2$, $NO_2$, CO, $O_3$, and meteorological data
(air temperature and relative humidity) were obtained from the nearest station of the National Environ-
mental Monitoring Net sites (http://www.cnemc.cn/) for comparison. Figure 1 also illustrates the energy
consumption structure of industrial enterprises at the four cities, with YC, UR, LZ being energy produc-
tion cities, and XN being a heavy manufacturing city (Shan et al., 2018). The energy consumption data
for 2019 was obtained from the Statistical Yearbook sharing platform (https://www.yearbookchina.com/).

2.2. Chemical analysis

The chemical components of the samples were analyzed using multiple instruments. Firstly, a piece of
each filter (0.5024 cm$^2$) was analyzed for organic carbon (OC) and elemental carbon (EC) contents using
a Thermal/Optical carbon analyzer (DRI Model 2015A, Desert Research Institute, USA) with the
IMPROVE-A method (Chow et al., 2007). One-quarter of the YC filter and one-half of the other three
city filters were extracted in 30ml Milli-Q water (18.2 M Ω·cm) using an ultrasonic bath for 40 minutes.
To minimize chemical reactions and evaporation loss during sonication, ice was added to the ultrasonic
bath. Water-insoluble residuals were eliminated by filtering extracts via a 0.45-μm PTFE syringe filter





(Pall Life Sciences, USA). The concentrations of water-soluble inorganic ions (WSIs) (Cl⁻, NO₃⁻, SO₄²⁻,
NH₄⁺, Na⁺, K⁺, Ca²⁺, and Mg²⁺) in the water extracts were analyzed using two 881 compact ion chroma-
tography systems (Metrohm, Herisau, Switzerland). The instrument and operation details can be found
elsewhere (Xu et al., 2015). WSOC was analyzed using a TOC analyzer (Elementar Vario TOC cube,
Hanau, Germany) with the method of total inorganic carbon (TIC) subtracted from total carbon (TC) (Xu
et al., 2015; Zhang et al., 2017).

2.3. Analysis by ultraviolet-visible absorption spectroscopy

Absorption spectra of water extracts in the 200–700 nm wavelength range were obtained using a UV-
visible absorption spectrophotometer (UV-2700; Shimadzu, Kyoto, Japan) at 1 nm intervals with Milli-
Q water as the reference. The absorption spectra were corrected by subtracting the absorbance at each
wavelength from the average absorption values between 690–700 nm (A₇₀₀).

The light absorption coefficient at a given wavelength (Abs$_\lambda$) of water extracts was calculated as follows:
$$Abs_\lambda = (A_\lambda - A_{700}) \cdot \frac{V_l}{V_a \cdot l} \cdot \ln(10) \tag{1}$$
where A$_\lambda$ is the absorbance at wavelength $\lambda$; V$_l$ is the extract volume (30 mL), V$_a$ is the volume of air
passing through the filter, and l is the optical path length, 1 cm.

The mass absorption efficiency (MAE$_\lambda$ m²/g ) of water extracts and their wavelength dependence of
light absorption can be derived as follows:
$$MAE_\lambda = \frac{Abs_\lambda}{C_{WSOC}} \tag{2}$$
$$MAE_\lambda = K \cdot \lambda^{-AAE} \ (330\text{nm} \leq \lambda \leq 400\text{nm}) \tag{3}$$
where $K$ is a constant related to light absorption, and $C_{WSOC}$ is the mass concentration of WSOC (mg/L)
in the extract. For simplicity, the absorption at 360–370 nm (mean 365 nm) was used to characterize the
absorption of BrC (Hecobian et al., 2010).

The radiative forcing of WS-BrC relative to EC ($f_{BrC}$) was estimated as follows (Kirillova et al., 2014):



$$f_{BrC} = \frac{\int I_{(\lambda)} \cdot \left\{ 1 - e^{-\left( MAE_{WSOC,365} \cdot \left(\frac{365}{\lambda}\right)^{AAE_{WSOC}} \cdot [WSOC] \cdot h_{ABL} \right)} \right\} d\lambda}{\int I_{(\lambda)} \cdot \left\{ 1 - e^{-\left( MAE_{EC,550} \cdot \left(\frac{550}{\lambda}\right)^{AAE_{EC}} \cdot [EC] \cdot h_{ABL} \right)} \right\} d\lambda}$$
(4)

Where $[EC]$ was the mass concentration of EC (μg/m³); $I(\lambda)$ was the clear-sky Air Mass 1 Global Hor-
izontal (AM1GH) solar radiance spectrum; $MAE_{EC,550}$ set to 7.5 m²/g; $h_{ABL}$ refers to the atmospheric
boundary layer(1000m); the details are described elsewhere (Kirillova et al., 2014; Bosch et al., 2014).

2.4. EEM fluorescence spectra analysis

The three-dimensional excitation-emission matrix spectroscopy (EEMs) of the samples were obtained
using an F-7100 fluorescence spectrometer (Hitachi High-Technologies, Japan). The EEMs were meas-
ured in the range of 200 to 400 nm with 5 nm intervals for excitation and 250 to 550 nm with 1 nm
intervals for emission. The inner filter effect was reduced by diluting the extracts until their absorbance
was below 0.1 at 254 nm (Ohno, 2002). The original EEMs were processed following : (1) subtracting
the Milli-Q water spectrum to reduce background influence; (2) interpolating to eliminate the interfer-
ence signals of the Rayleigh scattered light; (3) adjusting the fluorescence intensity by dividing the Ra-
man peak area of Milli-Q water at Ex=350 nm to remove instrument dependency (Lawaetz and Stedmon,

179   2009).


The resulting fluorescent intensity (unit of RU) was further processed using parallel factor analysis
(PARAFAC) to group potential similar chemical components. This was done using the MATLAB 2016b
software combined with the DOMFluor and drEEM toolboxes (Murphy et al., 2013). The 6-component
model was eventually chosen from 2–10 component PARAFAC models because the residual errors de-
creased markedly when the number of components increased from 2 to 6 (Figure S1). In addition, the 6-
component model has sensible spectrums of fluorescent components.

Furthermore, the fluorescence properties of the water extracts were examined through the fluorescence
indices. The humification index (HIX) was calculated by the ratio of the integrated fluorescence emission
intensity in the region of 435–480 nm to 300–345 nm under the excitation wavelength of 255 nm. The



biological index (BIX) was calculated by the ratio of the emission intensity of 380 nm to 430 nm under
the excitation wavelength of 310 nm (Zsolnay et al., 1999; Mcknight et al., 2001; Yang et al., 2020b).
The average fluorescence intensities (AFI) were calculated in the region of 200–400 nm for excitation
wavelengths and 250–550 nm for emission wavelengths.

The apparent fluorescence quantum yields (AQY) were calculated as follows:
$$AQY_\lambda = \frac{\int FI(\lambda_{Ex}, \lambda_{Em})d\lambda_{Em}}{UVA(\lambda_{Ex})\int d\lambda_{Em}}\Big|_{Ex} \qquad (5)$$

Where FI represents the fluorescence intensity (RU) at each excitation (200–400 nm) and emission (250–
550 nm) wavelength.

2.5. pH titration experiment

To investigate the variation of light-absorbing and fluorescent properties of ambient aerosols under the
influence of pH, we selected samples with higher WSOA concentrations at each site and recorded their
UV-vis absorption and fluorescence spectra at different pH settings. We adjusted the water extracts to pH
2 using 2 M HCl and then titrated with 0.1 M NaOH to different pH values until the pH reached 12, as
measured by a pH meter (Orion Star A111, Thermo Fisher Scientific, Waltham, MA, USA). We calibrated
the pH meter with buffer solutions of pH 4.01, 7.00, and 10.01 during the measurement.

2.6. HR-AMS Measurements

Offline high-resolution time-of-flight aerosol mass spectrometry (HR-AMS) was used to analyze the ion-
group, elemental ration, and oxidation properties of water-soluble organic aerosol (WSOA). Each water
extract was atomized with argon (purity: 99.9999%) to eliminate interference from $CO_2$ in the air. The
generated aerosol was passed through a silica gel diffusion dryer to remove water vapors, and the aerosol
particles were then sampled into an HR-AMS instrument (Aerodyne Inc., Billerica, MA, USA) through
an aerodynamic lens inlet. The HR-AMS was operated in V- and W-mode. After each sample, deionized
water samples were used to clean the sampling line and analyzed as the blank sample using the same
procedures. Elemental ratios, including O/C, H/C, N/C, and OM/OC, were determined based on high-





resolution mass spectra (m/z up to 120) and the Improved-Aiken (I-A) method (Canagaratna et al.,
2015). The elemental contributions of C, O, H, and N reported are mass-based, and more details can be
found elsewhere (Xu et al., 2013). Positive matrix factorization (PMF) was performed on high-resolution
mass spectra of organics at four cities to identify the potential source factors of WSOA, and four factors
were ultimately decomposed. A detailed description of PMF can be found in Zhao et al. (2022).

In addition, the mass concentration of WSOA were calculated as follows:
$$WSOA = \mathrm{WSOC} \cdot \mathrm{OM/OC} \qquad\qquad (6)$$

**3. Results and discussions**
3.1. Overview of the field observations

Figure 2 presents an overview of the time series of meteorological conditions (air temperature, relative
humidity (RH), and precipitation), $Abs_{365}$, AFI, as well as the mass concentrations of WSIs, WSOA, and
EC in the four cities. The weather during the study was generally sunny, cold, and dry (Figure 2a). For
instance, the average daily air temperature was $-3.2 \pm 3.4$ °C at YC, $-4.4 \pm 2.2$ °C at XN, $-9.2 \pm 3.7$ °C
at UR, and $-3.8 \pm 2.5$ at LZ, and the average RH was $62.4 \pm 15.8\%$ at YC, $56.1 \pm 14.7\%$ at XN, and $58.1$
$\pm 9.7\%$ at LZ. At UR, there were relatively higher RH condition ($83.9 \pm 6.6\%$), frequent foggy weather,
and two significant snowfall events, mainly due to the cold and wet air mass from Arctic Ocean during
winter (Yang et al., 2020a). Both the cold/wet and cold/dry weather conditions in our study facilitated
the study on different chemical processes.

The mass concentrations of chemical species in $PM_{2.5}$, as well as their mass fractions of all species varied
dynamically during the sampling period in four cities (Figure 2c and 2d). Heavy pollution (defined as
the daily average $PM_{2.5}$ mass concentration higher than 150 μg m$^{-3}$) occurred frequently in UR and fol-
lowed by YC (Figure 1). For example, the samples ID of 10–13 and 36–40, both occurred under meteor-
ological conditions with high RH condition (Figure 2a and 2c), which were favorable for the secondary
generation of atmospheric particulate matters (Sun et al., 2013). Therefore, the contribution of secondary





inorganic ions (SNA: sulfate + nitrate + ammonium) showed an important contribution to the recon-
structed $PM_{2.5}$ mass (WSOA + EC + WSIs) with an average value of 53.0 ± 12.7%, 41.6 ± 11.5%, 67.3
± 7.8%, and 41.8 ± 7.3%, respectively, in YC, XN, UR, and LZ, and could be as high as 65.6–76.9%
during heavy pollution period at YC and UR (Figure 2d). WSOA was also a major component of $PM_{2.5}$
accounting for 31.4 ± 5.5%, 40.6 ± 5.4%, 21.8 ± 4.6%, and 37.7 ± 4.1% at YC, XN, UR, and LZ, respec-
tively. The mass concentration of EC contributed comparably at each city (5.8–8.9%). The higher con-
tribution of SNA at YC and UR, in contrast with the higher contribution of WSOA at XN and LZ, likely
suggests their different sources and chemical processes. Comparing with the results in literatures, the
contributions of WSOA and SNA at XN and LZ in our study are comparable with those in Xi'an, another
megacity in Northwest China, during wintertime (Huang et al., 2014), while YC and UR are more com-
parable with those cities in East China, such as Beijing, Tianjin, and Jinan, where there are relatively wet
condition during wintertime (Lei et al., 2021; Zhang et al., 2021c; Dao et al., 2022). Specially, the chem-
ical composition of $PM_{2.5}$ at UR during heavy polluted periods is highly consistent with Beijing (Sun et
al., 2020), and the Yangtze River Delta cities during haze episodes (Ge et al., 2017; Ye et al., 2017; Sun
et al., 2022).

The ratio of WSOC/OC is commonly used to predict the potential contribution of secondary organic
aerosol to total organic aerosol (Psichoudaki and Pandis, 2013). Overall, WSOC showed a strong corre-
lation with OC ($R^2$ = 0.84) with a linear slope of 0.55 for all samples from four cities (Figure S3). The
slope values varied among the cities, with YC and UR having higher values (0.61 and 0.59) than XN and
LZ (0.54 and 0.52), suggesting a potentially higher secondary OA formation in YC and UR. The
WSOC/OC values in our study are within the range reported in other cities during winter, such as Xi'an
(0.50 and 0.53) (Zhang et al., 2018; Liu et al., 2020), Beijing (0.74) (Ni et al., 2022) and Guangzhou
(0.71) (Tao et al., 2022). Significant correlations among WSOC, OC, EC, $Cl^-$, $K^+$ and SNA were found
in four cities, indicating similar sources of primary and secondary species (Figure S2, Table S3). Fur-
thermore, the light-absorption and fluorescence parameters ($Abs_{365}$ and AFI) followed the variations of
WSOC, and we observed significant correlation between them ($R^2_{Abs\ Vs.\ WSOC}$ = 0.87; $R^2_{AFI\ Vs.\ WSOC}$ =
0.61). These findings suggest indicating an important contribution of WS-BrC chromophores to WSOC.



3.2. Bulk optical properties of WS-BrC

Figure 3a presents the average MAE spectra of WSOA from various cities. XN and LZ exhibited signif-
icantly higher $MAE_{365}$ values (on average $1.22 \pm 0.18$ and $1.19 \pm 0.12$ m²/g) compared to YC ($1.02 \pm$
$0.23$ m²/g) and UR ($0.78 \pm 0.16$ m²/g) (t-test, $P < 0.01$) (Table 1). The difference in $MAE_{365}$ values among
the cities could be attributed to the chemical composition and/or oxygenation state of BrC (Chen et al.,
2018; Chen et al., 2020b). When compared with previous studies in other typical Chinese cities during
wintertime, the $MAE_{365}$ values in our four cities were lower than those in Xi'an (1.33 and 1.50 m²/g)
(Huang et al., 2018; Yuan et al., 2020). However, the $MAE_{365}$ in XN and LZ were comparable to those
of Beijing (1.21–1.26) (Du et al., 2014; Cheng et al., 2016; Li et al., 2020b), while the $MAE_{365}$ in YC
were close to those of southern cities in China, such as Guangzhou ($0.93 \pm 0.06$ m²/g), Nanjing (1.04
m²/g) (Fan et al., 2016; Chen et al., 2018). Moreover, the $MAE_{365}$ in UR was similar to that of Yangzhou
($0.75 \pm 0.29$ m²/g) (Chen et al., 2020b).

AAE denotes the wavelength dependence of light absorption of BrC, an important optical parameter that
can be used to infer the chemical properties of BrC (Andreae and Gelencsér, 2006). In this study, we
found that the AAE values in YC, XN, and UR were $6.8 \pm 0.7$, $7.1 \pm 0.4$, and $6.9 \pm 0.3$, respectively. We
observed significantly lower AAE values ($6.4 \pm 0.5$) in LZ (t-test, $P < 0.01$) (Table 1). Our AAE values
fall within the range of values reported in other cities during winter for $PM_{2.5}$ water extracts calculated
in the same wavelength range, such as in Nanjing (6.8) (Chen et al., 2018), Beijing (7.3 and 7.5) (Du et
al., 2014; Cheng et al., 2016), and Guangzhou ($6.7 \pm 0.1$) (Fan et al., 2016). Previous studies have sug-
gested that higher AAE values may be associated with primary biomass combustion emissions and/or
SOA formation. For instance, Chen and Bond (2010) emphasized that particles from smoldering of var-
ious wood have largely values between 7–16. Lambe et al. (2013) indicated that secondary BrC generated
by laboratory also has a higher AAE value (5.2–8.8). Therefore, the differences in AAE values among
the four cities may indicate their different sources or/and chemical compositions.

Saleh (2020) proposed an optical-based classification of BrC using the space of $MAE_{405}$ vs. AAE, which
is linked with their physicochemical properties (i.e., molecular sizes and solubility) and atmospheric





aging (i.e., photo-enhancement or photo-bleaching). Almost all samples in this study fell into the region
of W-BrC, which is similar to ambient samples from other studies (Zhou et al., 2021; Xu et al.,
2022). However, a few samples at UR fell into the region of VW-BrC. Furthermore, the WS-BrC in YC
exhibited a broader range than other cities, indicating multiple sources and/or processes for WS-BrC in
this city. Higher AAE and $MAE_{405}$ values were found in XN, which could be associated with biomass
burning emissions. Additionally, WS-BrC in LZ was the closest to the region of M-BrC among the four
cities, and the different positions could be related to their chemical processes in each city. For instance,
upon examining the relationship between $MAE_{365}$ and the O/C ratio, a positive correlation was observed
only in LZ, while negative relationships were observed in other cities (Figure 10). The chemical
processes of WS-BrC are discussed in detail in section 3.7.

The radiative forcing of WS-BrC relative to EC in the wavelength range of 300−2500 nm ($f_{300-2500}$) was
found to be $7.05 \pm 2.61\%$ at LZ, followed by $5.91 \pm 2.54\%$ at XN, $5.10 \pm 2.33$ at YC, and $3.76 \pm 1.53$ at
UR. The relative radiative effect of WS-BrC in the ultraviolet wavelengths ($f_{300-400}$) was also calculated,
accounting for $32.78 \pm 7.95\%$, $29.74 \pm 7.15\%$, $25.45 \pm 7.62\%$, and $18.66 \pm 6.17\%$ of EC for LZ, XN,
YC, UR, respectively (Table 1). These results indicate that the contribution of the light-absorbing effect
from WS-BrC cannot be ignored, particularly in the UV range. The radiative effects of WS-BrC in our
study were similar to those in Xi'an ($4.99 \pm 1.23\%$ for $f_{400-700}$ and $25.9 \pm 5.47\%$ for $f_{300-400}$) but lower
than those in eastern cities, such as Beijing ($10.7 \pm 3.0\%$ for $f_{300-2500}$ and $42.2 \pm 12.8\%$ for $f_{300-400}$) and
Tianjin ($13.5 \pm 4.1\%$ for $f_{280-4000}$ and $54.3 \pm 16.9\%$ for $f_{300-400}$) and Nanjing ($9.6\%$ for $f_{300-2500}$ and $39.7\%$
for $f_{300-400}$) (Yan et al., 2015; Liu et al., 2020; Xie et al., 2020; Deng et al., 2022). This difference may be
related to the higher light-absorbing effect from BC due to energy-intensive industries and central heating
in winter in northwestern cities.

3.3. Fluorescence Indices

Fluorescence indexes, such as HIX and BIX, have been used in recent years to study the source and
chemical processes of atmospheric organic aerosols (Lee et al., 2013; Fu et al., 2015; Qin et al., 2018;
Deng et al., 2022). The HIX indicates the humification of WSOA, and it increases significantly upon



aging (Lee et al., 2013; Fan et al., 2020). The BIX is broadly in contrast with HIX and is known as the
freshness index. A higher BIX value implies a higher contribution of freshly released organics, while a
lower value indicates greater degrees of aging (Lee et al., 2013; Wen et al., 2021).

Table 1 shows that the HIX and BIX values were $1.85 \pm 0.36$ and $1.28 \pm 0.14$ at UR, $1.32 \pm 0.23$ and
$1.48 \pm 0.11$ at YC, $1.29 \pm 0.27$ and $1.49 \pm 0.15$ at XN, and $1.16 \pm 0.18$ and $1.52 \pm 0.11$ at LZ, respectively.
The highest HIX value and lowest BIX value are found in UR, indicating a higher degree of
aging/oxidation of WS-BrC. On the other hand, the lower HIX and higher BIX values observed in LZ
suggest a high contribution of freshly emitted BrC. These results are consistent with the results of the
MAE$_{365}$ discussed earlier.

The HIX displays a significant negative relationship with BIX for all the data ($R^2 = 0.86$, slope = 2.19)
(Figure 4a). Figure 4b shows a comparison of our results with other datasets from laboratory or ambient
aerosols in different cities in China. All the datasets can be roughly grouped into three zones colored by
the grey, pink, and blue dashed boxes, respectively. The freshly introduced materials generated from the
laboratory (gray box) are located in a much lower position than those of ambient samples. Differences
also exist in ambient samples with our samples (pink box) generally having higher (lower) BIX (HIX)
values than those from Eastern China (blue box) (Qin et al., 2018; Yue et al., 2019; Wen et al., 2021;
Deng et al., 2022), suggesting that our samples are generally less aged than those from Eastern cities,
although the position of UR locates in the overlapped range between these two zones.

3.4. Influence of pH on optical properties

Recent studies have shown that the optical properties of BrC vary depending on pH, which is important
for modeling its climate-forcing effect, as the general assumption of neutral state for aerosol in models
may not be accurate (Phillips et al., 2017). We investigated the effect of pH on the absorption and
fluorescence spectra of WSOA in our samples (Figures 5 and 6). The absorption spectra showed a
significant increase in absorbance with the increasing pH values (from 2 to 10), with the integrated
absorbance (300–450 nm) increased by 66.6%, 55.2%, 43.4%, and 25.3% relative to the pH = 2 level,



respectively, in YC, LZ, XN, and UR (Figure S4). The MAE$_{365}$ also increased with increasing pH values
(slope = 0.03–0.07), while the AAE decreased (slope = –0.15 to –0.40) (Figure S5). Specifically, the light
absorption spectra in YC were the most sensitive to pH variation, followed by LZ, XN, and UR, based
on these two slopes. The variations of light absorbance at a function of pH have been observed by
previous studies and were attributed to the protonation/deprotonation of carboxyl/phenolic functional
groups and/or their variation of macromolecular conformation (Lin et al., 2017; Phillips et al., 2017; Xu
et al., 2020b; Qin et al., 2020; Qin et al., 2022a). The different sensitivity of WS-BrC to pH at our
sampling sites suggests variations in chemical compositions among them, which could be further
investigated through fluorescence spectra.

The EEM spectra variations upon pH showed the highest values of fluorescent intensity at pH = 2 and
tended to decrease with increasing pH. However, for the YC and UR samples, the fluorescent intensity
at pH = 10 slightly increased compared to that at pH = 7 (Figure 6). The background mechanism of the
fluorescence variation on pH could be related to the rigid properties of fluorophores. The formation of
hydrogen bonds at low pH could give special chemical aggregates a stronger rigid planar conformation
and enhance fluorescence efficiency (Ghosh and Schnitzer, 1981; Mei et al., 2009). As the pH value
increases, the resulting anions repel each other with hydrogen bonds being broken, leading to a more
open conformation. This increase in conformational flexibility enhances light absorption but depresses
fluorescence. These explains why the light absorbance of WSOC increases under basic conditions, while
the fluorescence intensity increases under acidic conditions. In addition to being influenced by
conformation, the change in fluorescence spectra of chemical complexes with pH can also result from
charge transfer from special acidic/basic groups (Phillips and Smith, 2015; Phillips et al., 2017; Qin et
al., 2020). For the results in YC and UR (Figure 7), the fluorescence intensity spectra showing turning
points at pH2–4 and pH7–10, could be related to groups of –COOH and–NH$_2$ and/or –OH, respectively
(Cox et al., 1999; Milne et al., 2001; Phillips et al., 2017).

To identify the potential dominat chemical components of WS-BrC responsible for the pH-dependent
variations, the fluorescence peaks in EEM spectra related to special chemical compositions were
analyzed for their variations at different pH values (Chen et al., 2003; Fellman et al., 2010). These peaks



including peak A ( Ex/Em = 225–250/356–440 nm), classified as humic-like fluorophores (Fu et al., 2015;
Qin et al., 2018), peak T (Ex/Em = 270–280/330–355 nm), and peak B (Ex/Em = 270–280/290–310 nm),
classified as protein-like fluorophores (Chen et al., 2003; Birdwell and Engel, 2010), and peak M (Ex/Em
= 310–320/380–420 nm), categorized as oxygenated organic substances (Chen et al., 2003; Qin et al.,
2022b). Overall, peak A dominated the variation, contributing 78.5%, 69.1%, 74.1%, and 61.2% of the
total variation of all fluorescence peaks in YC, XN, UR, and LZ, respectively. Other peaks showed
moderate variations in the four cities, ranging from 8.3% to 12.4% for peak M, 11.2% to 17.9% for peak
T, and 1.6% to 7.6% for peak B. The variation trend of peak A was highly consistent with the trends of
the average fluorescence efficiency (AFI/TOC) and the average apparent quantum yield (AQY) over the
entire Ex/Em range at each city (Figures 7 and S7). These results suggest that the major fluorephores in
all the samples are humic-like compounds. Note that although the dominated contribution is from humic-
like compound (Peak A) in all the samples, the chemcial composition of humic-like compounds among
the cities are somewhat different, as shown by the different AQY peak shapes of this peak, which can be
further decomposed by PARAFAC model (Figure S6).

3.5. Fluorescent Components

Using the PARAFAC model, we were able to identify six chromophore components (C1–C6) from EEMs
(Figure 8 and Table S2). C1 is characterized by a primary peak (Ex/Em) at ~230 nm/375 nm and a
secondary peak of ~320 nm/375 nm. C5 also shows two similar peaks, but shifted to a shorter wavelength
at Ex (210/280 nm) and Em (373 nm). These two chromophores are characterized as less oxygenated
humic-like components (LO-HULIS) (Chen et al., 2016; Chen et al., 2021), with different oxidation
states between them, and C5 was likely from primary sources such as coal burning and vehicle emissions,
while C1 was secondary production. C2 shows a fluorescence peak (255nm/364 nm) and has been
observed in previous studies on fossil burning aerosol but has not been defined (Tang et al., 2020; Chen
et al., 2020a). C3, peaking at 240 nm/300 nm (Ex) and 414 nm (Em), is regarded as a highly oxygenated
humic-like chromophore (HO-HULIS), commonly considered as a secondary formation (Chen et al.,
2016; Yan and Kim, 2017; Cao et al., 2021). In particular, Hawkins et al. (2016) and Aiona et al. (2017)



found that the fluorescence generated by the aqueous-phase reaction of aldehydes with ammonium sul-
fate or amines highly matched the HULIS fluorescence peak (Ex < 250/~300 nm, Em > 400 nm) in
WSOA of ambient aerosol. C4 (Ex = 225/275 nm and Em = 338 nm) and C6 (Ex = 220 nm, Em = 292
nm) both peak at a short wavelength and are usually characterized as protein-like fluorophores (PLS)
(Yan and Kim, 2017; Wu et al., 2019; Chen et al., 2020a; Chen et al., 2021), but can also be phenol-like
substances or other aromatic compounds, especially for urban ambient aerosol samples (Barsotti et al.,
2016; Chen et al., 2020a; Cao et al., 2021; Deng et al., 2022). The averaged relative contributions of
chromophores are dominated by HULIS chromophores (C1, C3, and C5) with a total contribution of
56.5–68.4%, followed by PLS chromophores (C4 and C6) (16.5–22.3%), and then the undefined
chromophores (C2) (14.9–20.8%) (Figure 2e). Specifically, there were significant differences in the
relative content of each fluorescent component in four cities. For example, the content of C1 was higher
in YC (38.4% vs. 28.7–31.0% in the other three cities) (t-test, $P < 0.01$), the contents of C2 and C4 were
higher in LZ (20.8% and 21.1%) than in other three cities (14.7–16.2% and 11.2–18.6%) (t-test, $P < 0.01$),
and the content of C3 was significantly higher in UR than in YC, XN and LZ (28.6% vs. 18.8–19.4%)
(t-test, $P < 0.01$).

3.6. Source apportionment of WSOA by PMF analysis

Four WSOA factors were identified through PMF analysis on the high-resolution mass spectra of WSOA
at four cities, including a water-soluble primary OA (WS-POA), two highly oxidized oxygenated OA
(HO-OOA1 and HO-OOA2), and a less oxidized oxygenated OA (LO-OOA) (Figure 9). The mass spec-
trum of WS-POA is dominated by $C_xH_y^+$ (51%) fragment ions, followed by $C_xH_yO_1^+$ (24%), $C_xH_yO_2^+$
(14%), $C_xH_yN_p^+$ (6%), $H_yO_1$ (4%), and $C_xH_yO_zN_p^+$ (1%). The WS-POA has the lowest O/C (0.47) and the
highest H/C (1.68) among the four factors, but its O/C is much higher than those of online measurement
decomposed POA (<0.1) (Xu et al., 2020a; Zhao et al., 2022). In addition to oxygen-containing ions, the
WS-POA presents a few characteristics similar to those of the online measurement decomposed POA,
such as relatively high m/z at 55 and 57 with the ratio of m/z 55/57 being 2.67, 60 (fraction of signal =
0.39%), and 115 (fraction of signal = 0.21%), which could be related with cooking, biomass burning,
and coal combustion, respectively. These results suggest that WS-POA factor in our study represents



mixed primary sources. The mass contribution of WS-POA was $26.2 \pm 19.1\%$, $42.9 \pm 15.2\%$, $30.7 \pm$
$10.2\%$, and $48.8 \pm 9.3\%$ in YC, XN, UR, and LZ, respectively.

The mass spectrum of LO-OOA also shows a pronounced signal at m/z 115 (signal fraction = 0.36%)
and its concentration is highly correlated with some signals of PAHs, such as $C_6H_3^+$, $C_7H_4^+$, $C_8H_5^+$, and
$C_9H_5^+$ (Figure 9), indicating that LO-OOA was associated with coal combustion-induced WSOA. How-
ever, LO-OOA has significantly higher $OS_c$ (-0.05 vs. -0.74 for LO-OOA and WS-POA, respectively),
lower $C_xH_y^+$ ions (33%), and higher oxygenated ions combined (57% in total), including $C_xH_yO_1^+$ (32%),
$C_xH_yO_2^+$ (18%), $H_yO_1$ (4%), and $C_xH_yO_zN_p^+$ (2%), compared to the primary factor. These imply that LO-
OOA may represent a low oxidation OOA associated with coal combustion-induced WSOA. The same
factor was also observed in water-soluble aerosol samples from Beijing during winter (Hu et al., 2020).
The mass contribution of LO-OOA was $25.2 \pm 15.3\%$, $10.9 \pm 3.3\%$, $6.4 \pm 2.3\%$, and $7.3 \pm 1.6\%$ in YC,
XN, UR, and LZ, respectively.

The mass spectrum of HO-OOA1 was characterized by a distinct signal at m/z 44, which accounted for
20.4% of the total signal and was mainly composed of $CO_2^+$ (94%). Additionally, HO-OOA1 exhibited a
high O/C value (0.97), indicating its high oxidation. HO-OOA1 was significantly correlated nitrate ($r =$
0.33, P < 0.01) and odd oxygen ($O_x = O_3 + NO_2$) which are the products of photochemical processes
(Figure S7), suggesting that HO-OOA1 was photochemically produced (Herndon et al., 2008; Ye et al.,
2017). The mass contribution of HO-OOA1 were $29.6 \pm 18.1\%$, $37.2 \pm 10.1\%$, $13.4 \pm 10.2\%$, and 38.3
$\pm 8.5\%$ in YC, XN, UR, and LZ, respectively. The HO-OOA2 exhibited comparable O/C with that of
HO-OOA1 (0.99 vs. 0.97), but a higher N/C ratio (0.094 vs. 0.041) and a stronger correlation with RH
and sulfate than HO-OOA1, suggesting its potential aqueous processing production (Sun et al., 2016;
Wang et al., 2021). Furthermore, HO-OOA2 exhibited a significant correlation with $CH_2O_2^+$ ($r = 0.48$, P
< 0.01), a typical fragment ion for glyoxal, which could be generated from ring-breaking in the aqueous-
phase oxidation of polycyclic aromatic hydrocarbons (Chhabra et al., 2010; Wang et al., 2020). The mass
contribution of HO-OOA2 were $19.0 \pm 5.8\%$, $8.9 \pm 6.2\%$, $49.5 \pm 7.2\%$, and $5.1 \pm 2.7\%$ in YC, XN, UR,
and LZ, respectively. Overall, the WSOA at LZ had the highest contribution of POA than other cites,
while UR had the highest contribution of HO-OOA2 and YC had the highest contribution of LO-OOA



(Figure 2).

To further assess the relative light absorption contributions of WSOA from different sources, the four
WSOA factors mentioned above were assigned to $Abs_{365}$ using the multiple linear regression model
(MLR), as described in the following equation:
$Abs_{365} = a \times [WS\text{-}POA] + b \times [LOOA] + c \times [HOOA\text{-}1] + d \times [HOOA\text{-}2]$
where [WS-POA], [LOOA], [HOOA-1], and [HOOA-2] are the mass concentrations of WSOA factors;
and a, b, c, and d represent regression coefficients ($m^2/g$), which represents the MAE value for each factor.
The reconstructed $Abs_{365}$ fits well with the measured $Abs_{365}$, and the slope is close to 1 (Figure S8),
indicating the effectiveness of the algorithm. The fitted MAE values were higher for WS-POA and LOOA,
with values of 1.34 and 1.33 ($m^2/g$), respectively (Table S4). These values are comparable to those of
FF-POA (1.35) and LOOA (1.24) factors previously determined during the winter in Beijing (Wang et
al., 2021). The lower MAE of HO-OOA1 (1.10 $m^2/g$) may be due to photo-bleaching in photochemical
process (Browne et al., 2019; Chen et al., 2020b; Zhang et al., 2021b). The lowest MAE value for HO-
OOA2 (0.58 $m^2/g$) suggests a strong photobleaching effect of the aqueous-phase oxidation process (Wang
et al., 2021). The average relative contributions of different WSOA factors to light absorption are shown
in Figure 9c. WS-POA was the most important WS-BrC in our study, contributing 60%, 51%, 46%, and
30% of $Abs_{365}$ in LZ, XN, UR, and YC, respectively. HO-OOA1 also plays an important role in the
absorption of WS-BrC, contributing nearly 30% except in UR (only 13%). Although HO-OOA2 has a
small MAE value, its high-quality contribution in UR (50%) makes it an important WS-BrC contributor
in UR as well (32%), while it was the least contributing factor to $Abs_{365}$, with only 2–11% in other three
cities. LO-OOA were an equally important contributors to $Abs_{365}$ as WS-POA, with a contribution of 31%
vs. 30% in YC, while in other cities, the contribution was around 10%.

3.7. Chemical transformations of chromophores

The light-absorption capacity of BrC is closely related to atmospheric aging processes and its chemical
composition (Lin et al., 2016; Jiang et al., 2022). To understand the relationship between light-absorption
properties and chemical processes, several indexes, including O/C, H/C, N/C, and carbon oxidation state



(OSc), were investigated for the relationship with $MAE_{365}$ and AAE (Figure 10). The indexes of O/C and
OSc, which reflect the degree of atmospheric aging, show lower values at LZ than in other cities (on
average, 0.58 in LZ vs. 0.64–0.77 in other cities). A significant positive relationship was observed be-
tween O/C and $MAE_{365}$ in LZ ($r = 0.57$), while significant negative relationships were observed in YC
and UR ($r = -0.70$ and $-0.53$). These results suggest that photo-enhancement phenomenon occurred in
LZ, while photo-bleaching phenomenon occurred in other three cities. Fitting $MAE_{365}$ vs. O/C for all the
data of four cities by least-squares linear regression (Figure S9), the fitting trend turns when O/C is about
0.64 ($MAE_{365} = 1.04 \times O/C + 0.58$, when O/C<0.64; $MAE_{365} = -1.23 \times O/C + 1.90$, when O/C$\geq$0.64). A sim-
ilar phenomenon was found by Jiang et al. (2022) in different polar solvent extracts of Beijing $PM_{2.5}$
filter samples, which suggests that chemical processing is dominated by functionalization for low O/C
period, while it was mainly dominated by fragmentation for high O/C period. Therefore, we infer that
the photo-enhancement phenomenon in LZ probably due to the initial aging of fresh WSOA.

The variation between $MAE_{365}$ and H/C was broadly opposite to that of O/C. A significant positive cor-
relation between $MAE_{365}$ and H/C was observed in YC and UR, suggesting higher $MAE_{365}$ for fresh
aerosol. $MAE_{365}$ values showed significantly positive correlation with N/C in YC, XN, and LZ ($r = 0.57$,
0.50, 0.51, respectively), while weak correlation in UR  ($r = 0.11$), indicating that N-containing organic
species are effectively light-absorbing chromophores (Chen et al., 2018). To elucidate the potential chem-
ical composition and sources of N-containing species, a correlation analysis between $MAE_{365}$ and 65
fitted N-containing fragment ions was conducted. Higher correlation efficiencies were found for $C_xH_yN_p^+$
and $C_xH_yO_zN_p^+$ family ions, such as $CHN^+$, $CH_4N^+$, $C_2H_3N^+$, $C_2H_6N^+$, $C_3H_8N^+$, $C_2HNO^+$, $C_2H_2NO^+$,
$C_2H_5NO^+$, and $C_3H_4NO^+$ (Figure S10). These ions may be associated with amine, imine, N-heterocyclic
(e.g. imidazole), organonitrates, and nitroaromatic compounds, which have been proven to be important
BrC components in ambient aerosol (Farmer et al., 2010; Sun et al., 2012; Kim et al., 2019;
Kasthuriarachchi et al., 2020; Ditto et al., 2022; Jiang et al., 2022).

The relationship between AAE and O/C was also investigated. Firstly, in YC, AAE increases significantly
with increasing O/C (slope= 2.62, $r = 0.55$), which is likely related to highly chemical oxidation and the
formation of weakly light-absorbing O-containing functional groups (Sumlin et al., 2017; Zeng et al.,



2021), leading to a shift in the absorption spectrum towards UV wavelength (AAE increased) (Zhang et
al., 2013; Mo et al., 2018). However, a significant negative relationship between AAE and O/C was
observed in LZ (slope = −1.41, $r$ = −0.51). As mentioned above, the initial oxidation occurring in LZ
was dominated by functioniztion, which introduces functional groups to form auxochrome or
chromophore, leading to absorption enhancement and an absorption spectrum red-shift (AAE decreased)
(Lin et al., 2015; Zeng et al., 2021; Jiang et al., 2022). Additionally, the AAE values show a roughly
decreasing trend with increasing N/C in the four cities, which maybe due to the increase in the abundance
of N-heteroatoms leading to a red shift in the absorption spectrum (Jiang et al., 2022).

Crossing-correlation among fluorescent chromophores and chemical components of $PM_{2.5}$ were per-
formed to infer the possible sources and atmospheric chemical processes of WS-BrC (Figure 11a). The
results showed that C1, C3, and C6 are tightly correlated with SNA, especially sulfate, suggesting
secondary production sources. Meanwhile, C2, C4, and C5 are tightly correlated with primary species,
such as EC, $K^+$, and $Cl^-$, indicating primary emission sources. Furthermore, the fluorescent chromophores
were assigned to different WSOA factors based on their correlation. For example, C2, C4, and C5, which
are primary sourced chromphores, were significantly correlated with WS-POA. C1, characterized as less
oxygenated humic-like components, is significantly correlated with LO-OOA, representing secondary
chromophore components with less oxidation. C3, characterized as highly oxygenated humic-like
components, is significantly correlated with HO-OOA2, while a significant positive correlation is also
observed between C3 and HO-OOA1 when UR data were excluded (Figure S11). Thus C3 was likely
produced by aqueous-phase oxidation or photochemical oxidation, with different contributions from
these two aging pathways in different cities of our study. C6 showed significant correlation with HO-
OOA2, suggesting that this chromophore is a potential aqueous product. Due to the highly overlapping
EEM spectra of C6 with phenol chromophores (Barsotti et al., 2016; Chen et al., 2020a) and the signifi-
cant correlation between C6 and HO-OOA2, we speculate that C6 may be a phenol-like chromophore
and an aqueous-phase oxidation product. Recent studies have shown that benzene and its derivatives,
commonly found in coal combustion emission, have also been observed in atmospheric waters such as
clouds and fog (Raja et al., 2009). Benzene reacts readily with hydroxyl radicals in aqueous phase ($k_{OH}$
= 4.7×10$^5$ μM$^{-1}$ min$^{-1}$), which is much faster than its reaction to other atmospheric radicals such as ozone



($k_{O_3} = 6.1 \times 10^{-6}$ μM$^{-1}$ min$^{-1}$) and nitrate radicals ($k_{NO_3} = 4.0 \times 10^{-1}$ μM$^{-1}$ min$^{-1}$) or photolysis in the gas phase
(Minakata et al., 2009). Thus, it is likely the atmospheric chemistry of benzenes is initiated under •OH
at aqueous-phase to form phenol (Borrás and Tortajada-Genaro, 2012; Heath et al., 2013; Faust et al.,

569    2017).


Combing the information above into a Van Krevelen plot (H:C vs. O:C), which is usually used to describe
the evolution of organic aerosols (Heald et al., 2010; Ng et al., 2011; Canagaratna et al., 2015),  it is
clearly shown that there is a tight relationship between the evolution of chemical processes and the light
absorbance of chromaphores. Note that each WSOA factor in the Van Krevelen plot is colored by its
fitted MAE$_{365}$ values  and sized by the contribution of each WSOA factor to Abs$_{365}$. The six PARAFAC
components are associated with different WSOA factors based on their correlation among each other.
Specifically, aging processes along the direction of increased OSc from fresh species (WS-POA) to LO-
OOA and HO-OOA were clearly observed to be associated with a significant photobleaching
phenomenon, as evidenced by a decrease/increase in MAE$_{365}$/AAE values, especially for the aqueous-
phase oxidation, which resulted in the lowest MAE$_{365}$ value (0.58 m$^2$/g) and the highest AAE value (7.18).
Therefore, the slope from WS-POA to each OOA components could be used to some extent to describe
this photobleching phenomeno, i.e., –0.91 for WS-POA & LO-OOA, –0.53 for WS-POA & HO-OOA1,
and –0.34 for WS-POA & HO-OOA2. A lower slope (closer to –1) could be related to the addition of
carboxylic acid functional groups, while higher slopes (such as –0.5) could be related with the addition
of alcohol/peroxide functional groups. Additionally, the slope for each data set of the sampling sites
varied from –1.01 at LZ to –0.89 at XN, –0.78 at UR, and –0.71 at YC, further indicating the different
chemcial processes at each site and their different optical properties.

Figure 11c further shows the EEM profiles of the six PARAFAC components (dashed lines plotted), the
locations of the strongest fluorescence peaks (Ex/Em) (circles C1-C6), and the compound categories to
which they belong. The possible origins and atmospheric chemical transformations of these chromo-
phores are further elucidated by correlating the PMF results. It was found that the division of highly
oxygenated species region and less oxygenated species region proposed by Chen et al. (2016) was very



consistent with our PMF results, while our results are further subdivided into three regions of fresh spe-
cies, less oxidized and highly oxidized species, with each of the three regions circled by different shades
of brown boxes in Figure 11c. Note that the chemical transformation of the loss of primary chromophores
(fresh species) and the generation of secondary chromophores (less oxidized and highly oxidized species)
can occur through either photochemical oxidation or aqueous-phase oxidation, with different contribu-
tions from the two aging pathways in different cities. Additionally, some chromophores formed from
high oxidation processes have short emission wavelengths, which were originally classified as PLS chro-
mophores, providing a reference for determining PLS sources and processes in future studies.

**4. Conclusions**

In this study, a comprehensive comparison of the optical properties, potential sources, and chemical pro-
cesses of WSOA was conducted using atmospheric aerosols collected from four typical cities in North-
west China, namely Yinchuan (YC), Xining (XN), Urumqi (UR), and Lanzhou (LZ). The main conclu-
sions and environmental implications are obtained as follows.

Firstly, the optical properties of WSOA were found to be influenced by its chemical composition. The
$MAE_{365}$, HIX, and BIX values of XN (1.24 $m^2$/g, 1.29, 1.49) and LZ (1.19 $m^2$/g, 1.16, 1.52) were higher,
lower, and higher than those of YC (1.07 $m^2$/g, 1.32, 1.48) and UR (0.78 $m^2$/g, 1.85, 1.28), which prob-
ably due to a higher contribution of fresh WSOA in XN and LZ and a higher degree of humification and
aging/oxidation of WSOA in YC and UR. Secondly, the optical properties of WSOA were found to be
influenced by pH variation. The integrated absorbance (300–450 nm) and $MAE_{365}$ increased monoton-
ically with increasing pH in all four cities. The impact of pH on fluorescence EEM was far more complex,
involving the rigidity and planarity of molecule structure and the protonating/deprotonating of electron-
absorbing (–COOH and –$NO_2$) and electron-donating (–$NH_2$ and –OH) groups connected to the fluoro-
phore nuclei. The WSOA in YC and LZ were found to be most sensitive to pH variation and showed
different trends, indicating that their chemical structures are rich in different types of acid/base functional
groups. Thirdly, the optical properties of WSOA were found to be changed with aging/oxidation pro-
cesses. Obvious photo-bleaching were observed in YC and UR, while photo-enhancement was in LZ,



reflecting the role of the initial aging (functionalization) and further highly oxidation (fragmentation) of
fresh WSOA on the optical properties of WS-BrC based on the analysis of optical properties and bulk
chemical characteristics. Finally, the analysis combining chromophores with WSOA factors can be used
to illustrate the chemical processes and optical variation by V-K plot and EEMs plot which is useful for
understanding the dominated chemical pathway at each city.

Overall, this study provides insight into the optical properties, sources, and chemical transformations of
WS-BrC, which will provide an important reference for future studies to determine the sources and pro-
cesses of atmospheric chromophores and further help to estimate the climatic effects of atmospheric
aerosols and control carbonaceous aerosol pollution.


**Data availability**
The data used in this study can be accessed on request from corresponding author.

**Author contributions**
JX designed the research and MZ, HW, LG and WL collected samples. MZ processed data, plotted the
figures, and wrote the manuscript when JX and XZ gave constructive discussion. LZ and WZ had an
active role in supporting the experimental work. All authors contributed to the discussions of the results
and refinement of the manuscript.

**Competing interests**
The authors declare that they have no conflict of interests.

**Acknowledgment**
This work was partially supported by the National Natural Science Foundation of China (41977189) and
the Key Laboratory of Cryospheric Sciences Scientific Research Foundation (SKLCS-ZZ-2023).

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



**Figures**

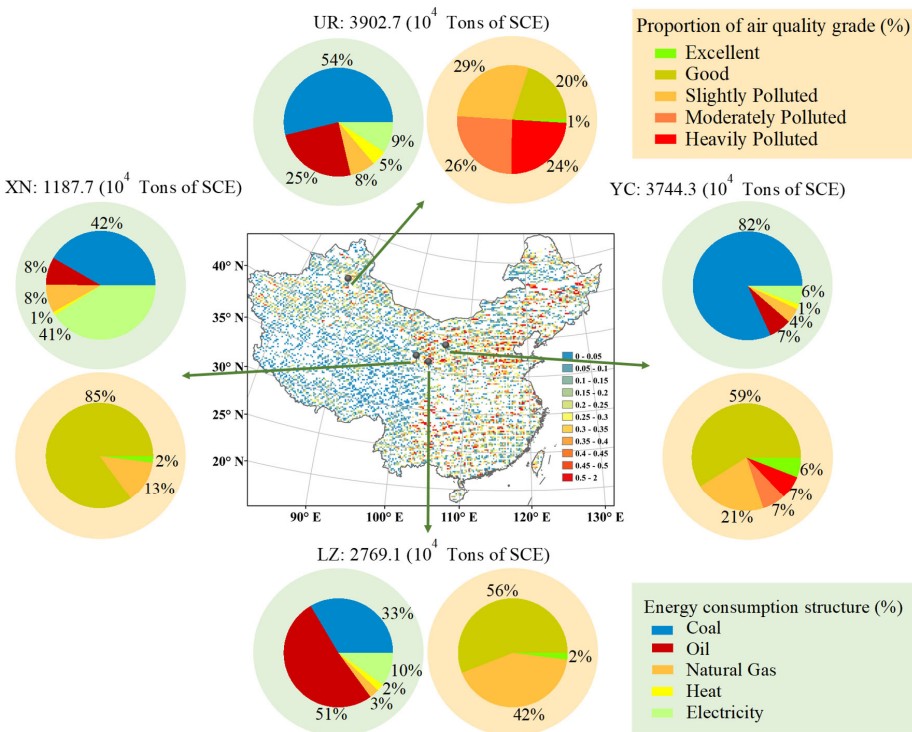

Figure 1. Location map for the sampling sites in this study, and the corresponding energy structure and air quality in each city during 2019. The spatial variation of $SO_2$ concentration in China is also shown at a resolution of 0.25°*0.25° retrieved from OMI satellite data during the sampling period of 2019/12-2020/1 (http://www.satdatafresh.com/). The mapping tool used was ArcGIS software. Pie charts around the map show the energy consumption structure during 2019 (left) (source: China Energy Statistical Yearbook) and the proportion of urban air quality grades during the sampling period (right) (https://www.zq12369.com/) at each city.

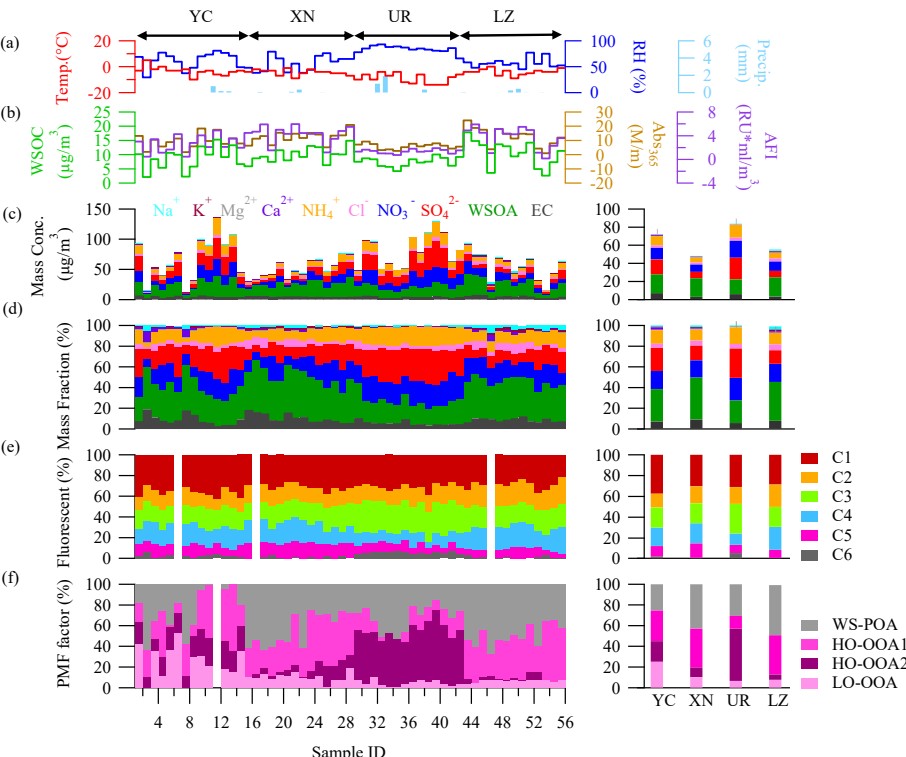

Figure 2. The combo plot for the results of aerosol samples versus sample IDs in this study. (a) Meteorological condition during the sampling including air temperature, relative humidity, and precipitation; (b) WSOC concentrations, the light-absorption (Abs), and average fluorescence intensity (AFI) of WSOA; (c) the concentrations of total identified species ( WSIs, OM, and EC); (d) The relative abundance of total identified species; (e) The relative abundances of the identified six fluorescent components by PARAFAC analysis; (f) Mass contributions of the factors resolved by PMF analysis on WSOA.

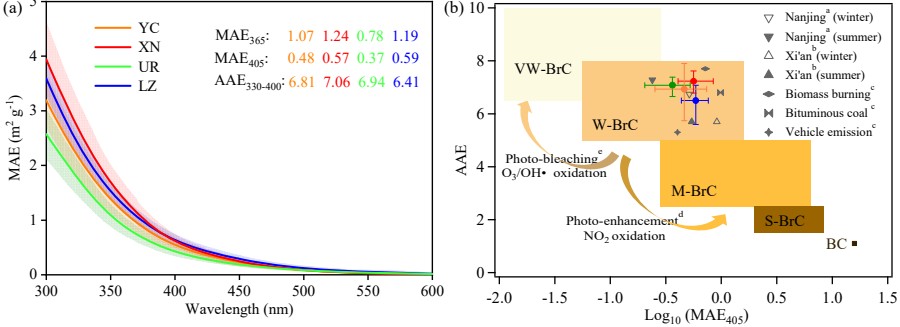

Figure 3. (a) The average MAE spectrum and their standard deviations of WS-BrC in each city represented by different colors. (b) Graphical representation of the optical-based BrC classes in the $\log_{10}(MAE_{405})$-AAE space (Saleh, 2020). The shaded areas respectively indicate very weakly (VW),



weakly (W), moderately (M), and strongly (S) absorbing BrC and absorbing BC. Grey marks indicate the data from previous studies about ambient BrC aerosol, i.e., [a](Chen et al., 2018) [b](Huang et al., 2018) [c](Tang et al., 2020). The curve with the arrowhead displays the variation tendency of optical properties of the lab-generated BrC aerosol and aged in the presence of $NO_3$ or $O_3$/OH radicals, i.e., [d](Li et al., 2020a) [e](Browne et al., 2019).

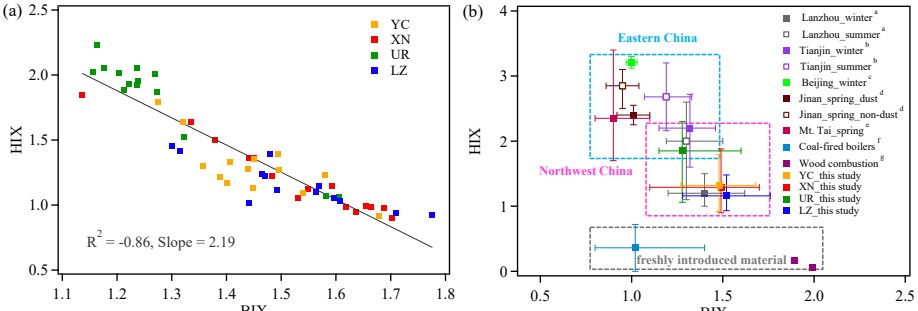

Figure 4. (a) Scatter plots of the humification index (HIX) as a function of the biological index (BIX) for WSOA in four cities. (b) Comparison plot of HIX versus BIX for this study and previous literatures, including WSOA from ambient aerosols in Lanzhou [a](Qin et al., 2018), Tianjin [b](Deng et al., 2022), Beijing [c](Qin et al., 2022b), Jinan [d](Wen et al., 2021), Mt. Tai [e](Yue et al., 2019), and from coal-fired aerosols [f](Yang et al., 2020b), and biomass burning aerosols [g](Fan, 2019).

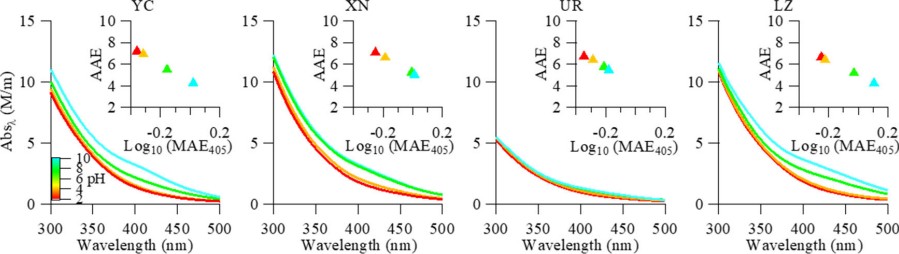

Figure 5. The Influence of pH on absorbance spectra (the insert figure shows the $\log_{10}(MAE_{405})$-AAE values at different pH).



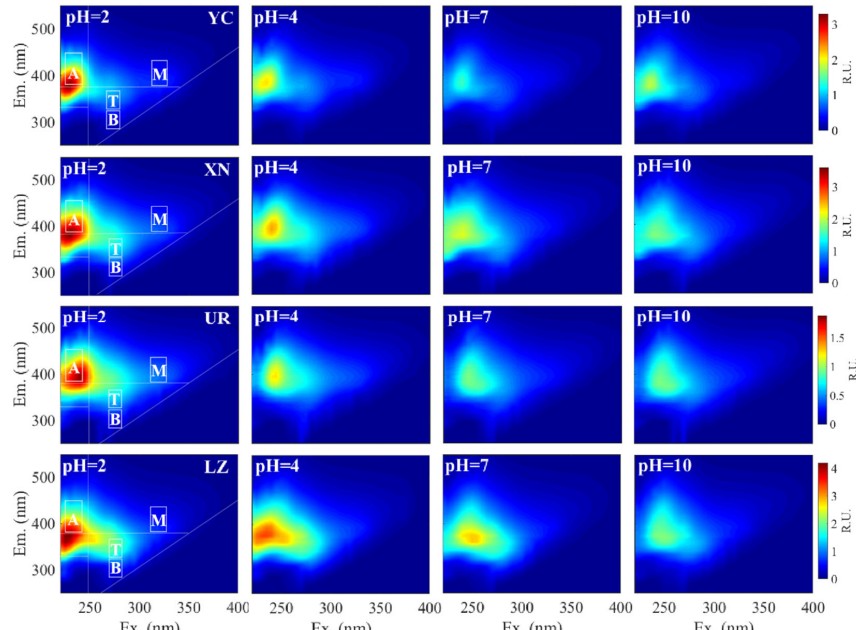

Figure 6. EEMs of WSOA at different pH values.

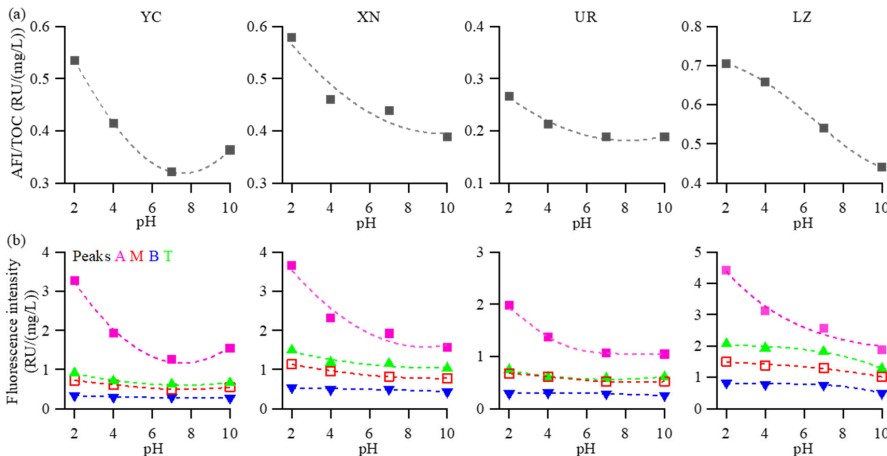

Figure 7. (a) The AFI/TOC and (b) maximum peak intensity of major fluorescence peaks as a function of pH values.



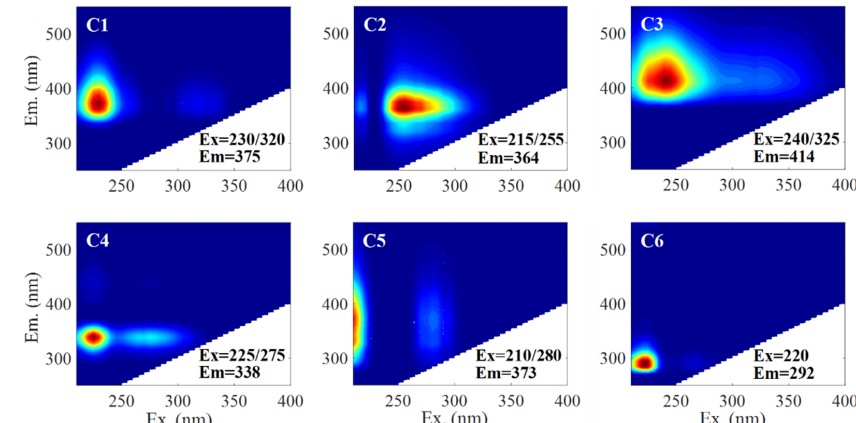

Figure 8. The EEM components identified by the PARAFAC model for the WSOA.



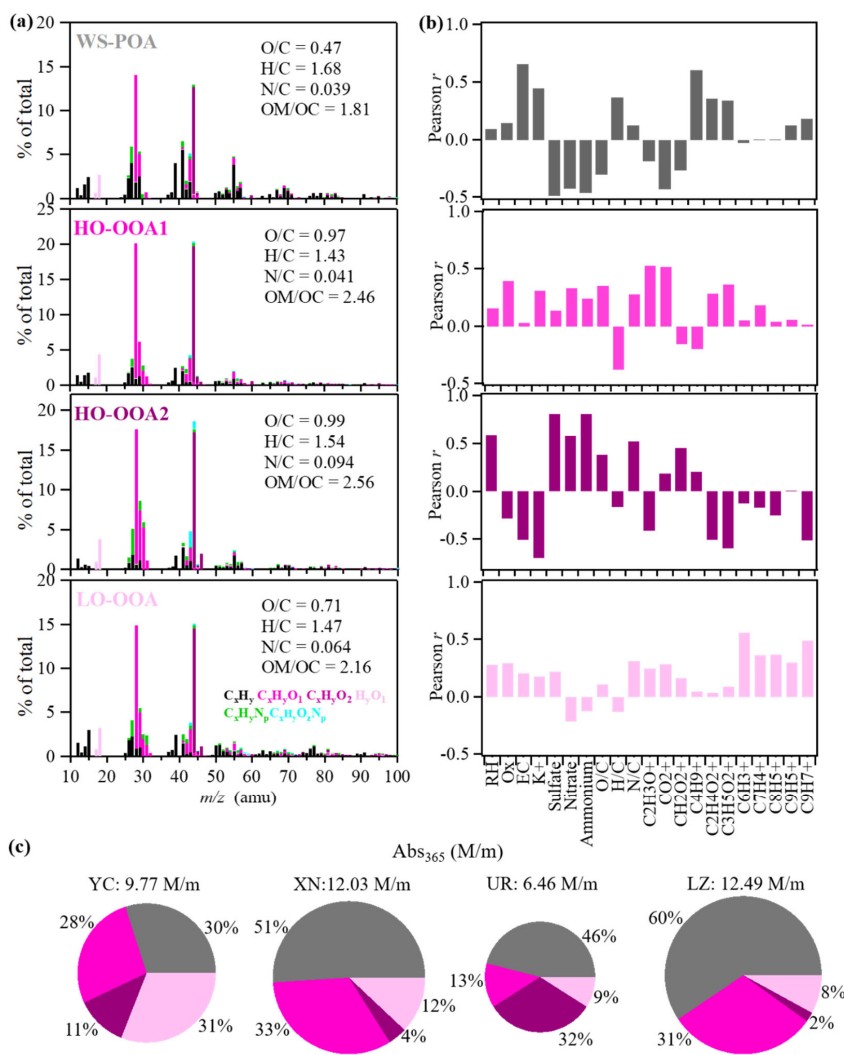

1128

Figure 9. (a) The mass spectra of PMF factors (WS-POA, HO-OOA1, HO-OOA2, LO-OOA), (b)
correlations between PMF factors and various tracers, and (c) average contributions of WSOA fac-
tors to light absorption at 365 nm.

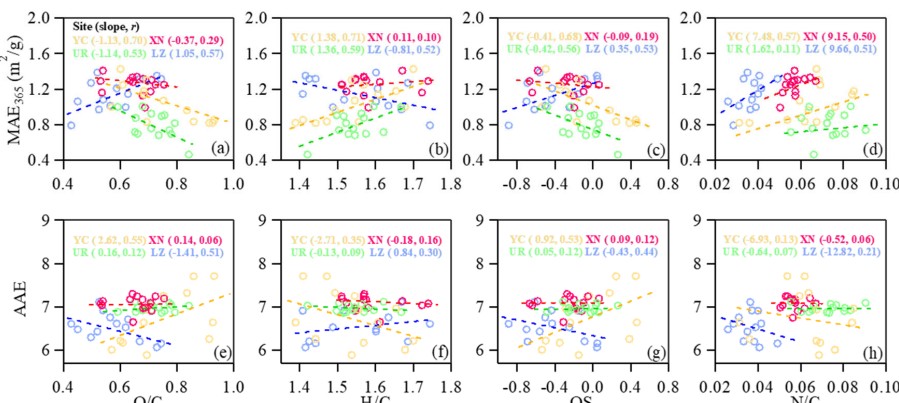

Figure 10. Scatter plots of MAE$_{365}$ (a–d) and AAE (e–h) vs. (a, e) O/C, (b, f) H/C, (c, g) OS$_c$, and (d, h) N/C in four cities. The slope and correlation coefficient ($r$) by fitting the scatter of each group are shown, and p-test is significant at the 0.05 level when $r$ reached 0.49.

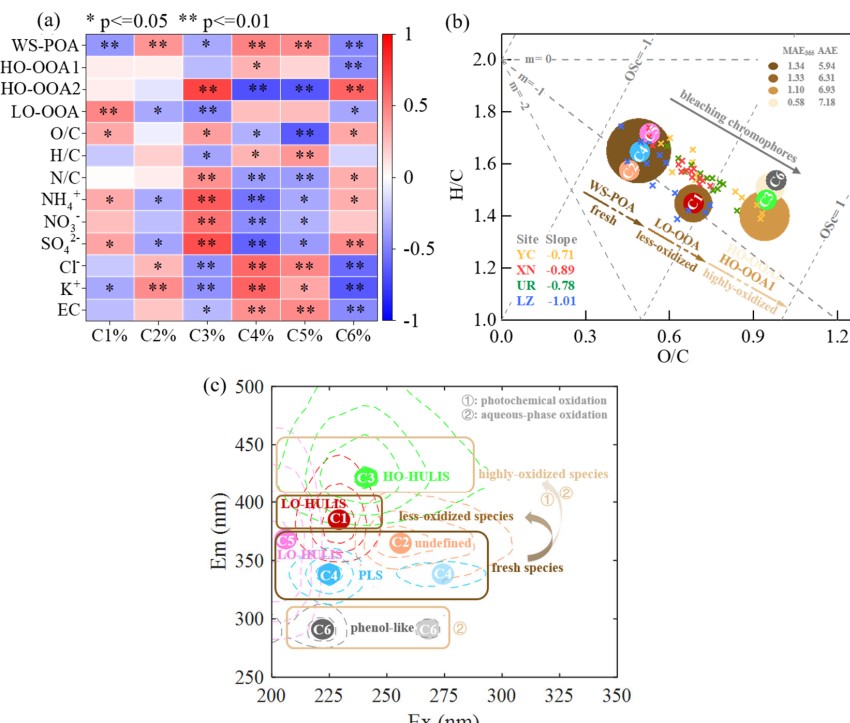

Figure 11. The diagram illustrate the aging from fresh species to less oxidized and/or highly-oxidized species and corresponding variation on optical properties. (a) Heatmap of the correlation analysis between the PARAFAC components and PMF factors, with highly significant correlations (p < 0.01) are marked by** and significant correlations (0.01 < p < 0.05) marked by *. (b) The Van Krevelen plot (H:C vs. O:C) for ambient WSOA samples and different WSOA factors in this study,



with the slopes of the fitted line for ambient WSOA samples from each city noted in the lower left.
The fitted $MAE_{365}$ and AAE values for each WSOA factor are noted in the upper right, and each
WSOA factor is colored by its $MAE_{365}$ value, with fading color indicating bleaching chromophores.
The size of the color block represents the average contribution of each WSOA factor to $Abs_{365}$.
Based on the correlation coefficients between PARAFAC components and PMF factors, C2, C4,
and C5 chromophores were assigned to WS-POA, C1 chromophore assigned to LO-OOA, and C3
and C6 chromophore assigned to HO-OOA (HO-OOA1 and HO-OOA2). (c) The Ex-Em plot for
fluorescence peak positions (Ex/Em) and the corresponding compounds of these six fluorophores.

**Tables**
Table 1. Light-absorbing properties of BrC in $PM_{2.5}$ water extract in four cities.

|  | YC | XN | UR | LZ |
|---|---|---|---|---|
| Light absorption property |  |  |  |  |
| AAE | $6.81 \pm 0.69$ | $7.06 \pm 0.44$ | $6.94 \pm 0.25$ | $6.41 \pm 0.51$ |
| $Abs_{365}$ (M/m) | $9.77 \pm 4.74$ | $12.03 \pm 3.16$ | $6.46 \pm 1.60$ | $12.49 \pm 4.94$ |
| $MAE_{365}$ ($m^2/g$) | $1.02 \pm 0.23$ | $1.22 \pm 0.18$ | $0.78 \pm 0.16$ | $1.19 \pm 0.12$ |
| $MAE_{405}$ ($m^2/g$) | $0.48 \pm 0.14$ | $0.57 \pm 0.11$ | $0.37 \pm 0.09$ | $0.59 \pm 0.10$ |
| $f_{300-400}$ (%) | $21.15 \pm 5.62$ | $22.47 \pm 5.85$ | $19.03 \pm 5.17$ | $25.17 \pm 6.5$ |
| $f_{300-600}$ (%) | $11.86 \pm 2.92$ | $12.44 \pm 3.04$ | $10.61 \pm 2.64$ | $14.11 \pm 3.61$ |
