# Peer review of "Brown carbon in fine particles in four typical cities in North- west China during wintertime: coupling optical properties with chemical processes"

_EGUsphere, 2023_

## Author Comment (AC2)

We appreciate the reviewers for their constructive comments and suggestions. The manuscript has been revised accordingly. Our point-by-point responses to the comments are presented below. The comments are in **black**, responses in **blue** and revised manuscript in **red**.

**Response to reviewer #1**

This manuscript presents an investigation of water-soluble brown carbon aerosols in several provincial cities in northwest China. These regions are distinctive due to their contrasting meteorological conditions and emission sources compared to "hotspots" such as the Beijing-Tianjin Hebei and Yangtze River Delta regions. Due to less exploration on aerosol chemistry in these cities, understanding of BrC aerosols in these urban areas has been limited so far. This manuscript presents many properties of BrC during wintertime. The most interesting point is that they provide a unique approach by combining BrC chromophores with WSOA factors, utilizing V-K diagrams and EEMs diagrams to illustrate chemical processes and optical changes. The results are useful for understanding of haze pollution and estimating of the climatic effects of BrC aerosols in energy-producing and heavy manufacturing cities in Northwest China. Overall, the manuscript fits well within the scope of ACP, and the results and presentation are reasonable and clear. However, there are a few issues that should be addressed before publication.

We thank the reviewer for his/her careful review and positive comments on the manuscript.

Major issues

This manuscript focuses on water-soluble brown carbon (WS-BrC); however, it is important to note that only a portion of BrC is water-soluble. Therefore, it is recommended to provide a more detailed description of water-soluble organic aerosols (WSOA) in the introduction section. Additionally, including relevant literature that discusses the molecular and chemical functional groups of WSOA would enhance the introduction section.

Agree. In the updated manuscript, we mentioned that WS-BrC is only a portion of BrC, accounting for approximately half of the total light absorption by solvent-extractable organic aerosols in urban areas, and exceeding half and even up to 90% in remote areas (line 88–93). Additionally, the information on the molecular and chemical functional groups of water-soluble BrC has been added, which has identified as major WS-BrC species, such as humic-like substances, phenols, nitroaromatics, and oxygenated aromatics (line 101–104).

"The solvent extracts from filter samples have been widely used to assess the absorbing properties of BrC (Zhang et al., 2017; Wu et al., 2019; Xu et al., 2020b; Zou et al.,

2023). Water-soluble BrC is only a portion of BrC, which accounts for approximately half of the total light absorption by solvent-extractable organic aerosols in urban areas (Cheng et al., 2017; Huang et al., 2018; Chen et al., 2020b), and exceeds half and even up to 90% in remote areas (Zhu et al., 2018; Li et al., 2022)."

"Recent studies have identified the major water-soluble BrC species, including humic-like substances, phenols, nitroaromatics, and oxygenated aromatics (Cai et al., 2020; Qin et al., 2022a; Jiang et al., 2022; Zhou et al., 2022)."

The evaluation of the radiative impact of WS-BrC is import for the radiative forcing of aerosol. In addition, the imaginary component of the complex refractive index of BrC is a critical input parameter for climate models. Numerous studies have employed the methodology proposed by Bikkina and Sarin (2019) to calculate the simple forcing efficiency (SFE) model of BrC to indicate the radiative forcing produced by a unit mass of PM on the atmosphere. These parameters could be considered.

Agree. The method for calculating the imaginary part ($k$) and the simple forcing efficiency (SFE) of WS-BrC are updated in the revised manuscript (line 191–205) as follows.

"The imaginary part $k$ of the particle refractive index represents the light absorption by aerosols and can be calculated as follows (Liu et al., 2013):

$$k_\lambda = \frac{MAE_\lambda \times \lambda \times \rho}{4\pi} \tag{4}$$

where $\rho$ (g/cm$^3$) is estimated as $\rho = (12 + H/C + 16 \times O/C) / (7 + 5 \times H/C + 4.15 \times O/C)$ (Kuwata et al., 2012).

The direct radiative forcing of WS-BrC is estimated by the simple forcing efficiency model (SFE). The wavelength-dependent $SFE_{Abs}$ (W/g) of WS-BrC is calculated as following equation (Bikkina and Sarin, 2019):

$$\frac{dSFE_{Abs}}{d\lambda} = D \frac{dS(\lambda)}{d\lambda} \tau_{atm}^2 (1-F_C) \times 2\alpha_S \times MAE_\lambda \tag{5}$$

where $S(\lambda)$ represents the wavelength-dependent solar irradiance, obtained from ASTM G173–03 reference spectra. Additionally, $\tau_{atm}$ refers to atmospheric transmission (0.79); D is assigned the value of 0.5, representing the proportion of daylight hours; $F_C$ is set to 0.6, representing the cloud proportion; $\alpha_s$ is fixed at 0.19 for the global average, representing the surface albedo (Chen and Bond, 2010)."

The results of $k$ and SFE are shown in Section 3.2 (line 340–349 and 383–391) as follows:

"In addition, the imaginary refractive index at 365 nm ($k_{365}$) for WS-BrC showed the same trend as $MAE_{365}$, i.e., XN (0.034 ± 0.007) > LZ (0.031 ± 0.007) > YC (0.029 ± 0.009) > UR (0.023 ± 0.007) (Table 1). Overall, the $k_{365}$ values in this study were

comparable to those reported in other urban areas, such as Kanpur in central India during winter (0.042) (Choudhary et al., 2021) and Jinan in northern China during spring (0.035) (Wen et al., 2021); however, they were higher than those reported in the Himalayan cryosphere during winter (0.009–0.026) (Choudhary et al., 2022) and in the northeast margin of Qinghai-Tibetan Plateau during summer (0.022) (Xu et al., 2020b). Previous studies indicated that photochemical aging decreased the $k$ values for BrC aerosols (Laskin et al., 2015; Sumlin et al., 2017)."

"We estimated the integrated mean SFE values of WS-BrC within the 300–700 nm range (SFE$_{300-700}$) due to the abrupt decrease of solar spectral energy below 300 nm and the negligible absorption of solar radiation by BrC above 700 nm. SFE$_{300-700}$ was larger at XN (4.42 ± 0.72 W/g) and LZ (4.35 ± 1.01W/g) than at YC (3.72 ± 0.90 W/g) and UR (2.97 ± 0.6 W/g) (Table 1), indicating their stronger light-absorbing capacity of WS-BrC. The SFE values in this study were similar to those in Hong Kong during winter (4.4 W/g) (Zhang et al., 2020), slightly higher than those in Jinan during spring (3.3 W/g) (Wen et al., 2021), but only half of those in laboratory biomass burning samples (7.7–8.3 W/g) (Lei et al., 2018)."

Minor issues

Line 38: Typo in "including three humic-like substances (LO-HULIS, HO-HULIS1, and HOHULIS2)". Only two factors in section 3.5.2.

Agree. We made a mistake here and revised it as follows.

"…including three humic-like substances (two LO-HULIS and one HO-HULIS)"

Line 112: Replace "PM2.5" with "PM$_{2.5}$".

Corrected.

Some formulas in the method section appear error italics and normal font.

Corrected.

Line 136: water-soluble inorganic ions Commonly abbreviated as WSIIs.

Corrected.

In Figure 2e and 2f, data is missing at different sample ids. What is the reason?

In Figure 2e, samples 6, 16, and 46 were excluded during the PARAFAC analysis due to their deviation on the loading and leverage. Similarly, sample 11 was excluded during the PMF analysis in Figure 2f.

Unify the spaces before and after the mathematical operator, e.g. Missing space before "=" in line178,535,566 and Missing space before "+" in line 484, 515.

Corrected.

In Figure 6, it can be observed that the fluorescence peak position shifts significantly with the change of pH, so it is suggested to briefly discuss it.

Agree. We have added the information on fluorescence peak at different pH values in Figure 6 as follows.

[Figure]

Figure 6. EEM spectra and the fluorescence peak (Ex, Em) of WSOA at different pH values in four cities.

The discussions on the variation of fluorescence peak with pH values are also presented in the revised manuscript (line 459–467), as follows.

"…, and the position variation of the fluorescence peak ($\lambda_{Em}$) can further reveal the types of acidic/basic groups (Qin et al., 2022a; Qin et al., 2022b). At XN and LZ, the $\lambda_{Em}$ redshifted at pH 2–4, indicating the deprotonation of electron-donating groups (e.g., −OH and −NH$_2$), while the $\lambda_{Em}$ blueshifted at pH 4–7, indicating the deprotonation of electron-withdrawing groups (e.g., −COOH, −C=O, and −NO$_2$). At LZ, the $\lambda_{Em}$ further redshifted at pH 7–10, suggesting another electron-donating group. In contrast, at YC

and UR, the $\lambda_{Em}$ redshifted in pH 2–7 and then blueshifted in pH 7–10, suggesting that the deprotonation of electron-donating and electron-withdrawing groups was different from that of LZ."

Line 416: Replace "255nm" with "255 nm"

Corrected.

Please unify to unit format, such as $m^2/g$ and $\mu g\ m^{-3}$

All units in the updated manuscript are consistently formatted.

Line 413-416: "C5 was likely from primary sources such as coal burning and vehicle emissions, while C1 was secondary production." Can you give a detailed explanation of how you came to this conclusion and please provide references.

In our findings, C1 and C5 chromophores were identified as less oxygenated humic-like substances (LO-HULIS), which are typically associated with combustion sources (Chen et al., 2016; Chen et al., 2020a; Chen et al., 2021). Furthermore, Matos et al. (2015) and Wang et al. (2020) noted that the emission peak of HULIS shifted to longer wavelengths likely due to the occurrence of chemical aging processes of those fluorophores. Therefore, in our study, C5 was identified as a primary chromophore, while C1 which the Ex/Em shifted slightly to longer wavelength was a secondary product.

We have rewritten this sentence in the revised version (line 494–501) as follows.

"C1 displayed a primary peak (Ex/Em) at 230 nm/375 nm and a secondary peak at 320 nm/375 nm, while C5 exhibited two similar peaks at 210 nm/373 nm and 280 nm/373 nm, albeit with a blueshift. These two chromophores were identified as less oxygenated humic-like substances (LO-HULIS), typically associated with combustion sources (Chen et al., 2016; Chen et al., 2020a; Chen et al., 2021). Therefore, we speculated that C5 is likely a primary chromophore, while a redshift of C1 suggests that C1 is a secondary product (Matos et al., 2015; Wang et al., 2020)."

What method is used to analyze the correlation between AMS and chromophores? Need to indicate the relative contribution or absolute strength of AMS factors or fragment ions used?

Agree. We conducted Pearson's correlation analysis between the relative abundance of WSOA factors and the relative content of fluorescent components. We have added this information to the revised manuscript (line 661–663) as follows.

"Furthermore, the fluorescent chromophores were assigned to different WSOA factors

based on Pearson's correlation analysis between the relative abundance of the four PMF factors and the relative content of the six PARAFAC components."

**Response to reviewer #2**

This study compared optical properties of BrC in wintertime aerosols in four capital cities of Northwest China. The authors also analyzed the chemical compositions by EEMs and HR-ToF-AMS, and discussed the difference in sources and chemical processes of BrC in the four cities. This study was well organized, and the paper was also well written. I recommend the manuscript to be accepted with a moderate revision. Detailed comments were listed below.

We greatly appreciate the reviewer for his/her positive comments and constructive suggestions.

Major comment:

In L16-19, and L86-94, the authors seem try to emphasis the important contribution of coal burning to the aerosols and BrC in the four capital cities of Northwest China. However, both optical properties and chemical compositions (such as AMS data) didn't support it. In particularly, in L297-302, and L310-311, biomass burning would be the dominant source of BrC in this study. I suggest to revised the Abstract and Introduction for this point.

We agree with the reviewers' comments. One of the primary objectives of this study is to investigate the optical properties of coal combustion emissions in these northwestern cities, which represent the dominant energy consumption source. In our findings, coal combustion emissions were indeed detected; however, the signals were generally weak, likely owing to their water-insoluble nature. The revised manuscript now weakens this emphasis in the abstract and introduction sections as follows (line 73–75 and line 108–112 in the manuscript).

Firstly, the following sentences in the abstract section are removed: "Most previous studies have predominantly focused on bulk optical properties of ambient BrC from biomass burning emitted primary or secondary BrC aerosol. Few studies have focused on fossil-fuel-influenced BrC aerosol, especially coal combustion emissions."

Secondly, we rephrase the content of the importance of coal energy in the introduction as follows.

"Coal combustion is another important source of primary BrC, particularly in urban areas during the winter heating period (Tan et al., 2016; Hu et al., 2020; Yuan et al., 2021; Deng et al., 2022)."

"Despite previous research on the chemical compositions and source apportionment of atmospheric aerosols (Xu et al., 2014; Xu et al., 2016; Tan et al., 2016; Xu et al., 2020a; Zhang et al., 2021), the current understanding of the optical properties and formation mechanisms of brown carbon over Northwest China is quite limited and deserves more attention."

Minor comments:

L269-271, I would say WSOC/OC in Beijing (0.74) and Guangzhou (0.71) were obviously higher than those in this study.

Agree. We made mistakes for these numbers which should be Beijing (0.54) and Guangzhou (0.55). The contents in the manuscript are revised accordingly.

"…, Beijing (0.54) (Ni et al., 2022), and Guangzhou (0.55) (Tao et al., 2022)."

L307 and 308, please give the full spelling of W-BrC and VW-BrC.

The full spelling of W-BrC, VW-BrC, and M-BrC are presented in the updated manuscript.

"…W-BrC (Weakly absorptive BrC) ..."
"…VW-BrC (Very weakly absorptive BrC) …"
"…M-BrC (Moderately absorptive BrC) …"

L361-363, Did't you mean the absorbance at pH=10 increased.... relative to the pH=2?

Yes. This sentence has been rephrased in the updated manuscript.

"The integrated absorbance (300–450 nm) increased by 66.6%, 55.2%, 43.4%, and 25.3% in YC, LZ, XN, and UR at pH 10 relative to pH 2 (Figure S4)."

L490, please give the full spelling of FF-POA.

The full spelling of FF-POA is presented in the updated manuscript.

[revised manuscript text omitted]